# Characterizing Bias in Classifiers using Generative Models

**Daniel McDuff, Yale Song and Ashish Kapoor**
Microsoft
Redmond, WA, USA
{damcduff,yalesong,akapoor}@microsoft.com

**Shuang Ma**
SUNY Buffalo
Buffalo, NY
shuangma@buffalo.edu

## Abstract

Models that are learned from real-world data are often biased because the data used to train them is biased. This can propagate systemic human biases that exist and ultimately lead to inequitable treatment of people, especially minorities. To characterize bias in learned classifiers, existing approaches rely on human oracles labeling real-world examples to identify the "blind spots" of the classifiers; these are ultimately limited due to the human labor required and the finite nature of existing image examples. We propose a simulation-based approach for interrogating classifiers using generative adversarial models in a systematic manner. We incorporate a progressive conditional generative model for synthesizing photo-realistic facial images and Bayesian Optimization for an efficient interrogation of independent facial image classification systems. We show how this approach can be used to efficiently characterize racial and gender biases in commercial systems.

## 1   Introduction

Models that are learned from found data (e.g., data scraped from the Internet) are often biased because these data sources are biased (Torralba & Efros, 2011). This can propagate systemic inequalities that exist in the real-world (Caliskan et al., 2017) and ultimately lead to unfair treatment of people. A model may perform poorly on populations that are minorities within the training set and present higher risks to them. For example, there is evidence of lower precision in pedestrian detection systems for people with darker skin tones (higher on the Fitzpatrick (1988) scale). This exposes one group to greater risk from self-driving/autonomous vehicles than another (Wilson et al., 2019). Other studies have revealed systematic biases in facial classification systems (Buolamwini, 2017; Buolamwini & Gebru, 2018), with the error rate of gender classification up to seven times larger on women than men and poorer on people with darker skin tones. Another study found that face recognition systems misidentify people with darker skin tones, women, and younger people at higher error rates (Klare et al., 2012). To exacerbate the negative effects of inequitable performance, there is evidence that African Americans are subjected to higher rates of facial recognition searches (Garvie, 2016). Commercial facial classification APIs are already deployed in consumer-facing systems and are being used by law enforcement. The combination of greater exposure to algorithms and a reduced precision in the results for certain demographic groups deserves urgent attention.

Many learned models exhibit bias as training datasets are limited in size and diversity. The "preferential selection of units for data analysis" leads to sample selection bias Bareinboim & Pearl (2016) and is the primary form of bias we consider here. Preferential selection need not necessarily be intentional. Let us take several benchmark datasets as exemplars. Almost 50% of the people featured in the widely used MS-CELEB-1M dataset (Guo et al., 2016) are from North America (USA and Canada) and Western Europe (UK and Germany), and over 75% are men. The demographic make up

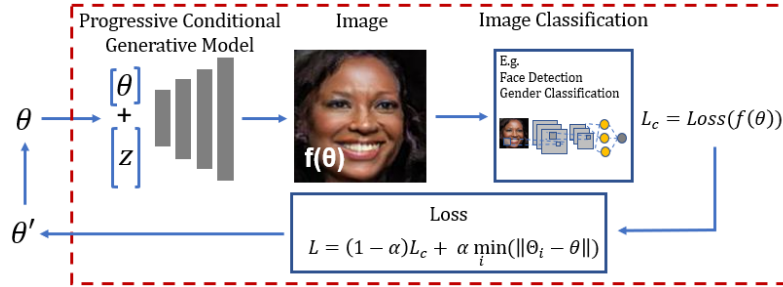

Figure 1: We propose a composite loss function, modeled as a Gaussian Process. It takes as input conditioning parameters for a validated progressive conditional generative model, and produces an image $f(\theta)$. This image is then used to interrogate an image classification system to compute a classification loss $L_c = Loss(f(\theta))$, where the loss is a binary classification loss (0 or 1), capturing whether the classifier performed accurately or poorly. This loss allows *exploitation* of failures and is combined with a diversity loss to promote *exploration* of the parameter space.

of these countries is predominantly Caucasian/white.[1] Another dataset of faces, IMDB-WIKI (Rothe et al., 2015), features 59.3% men and Americans are hugely over-represented (34.5%). Another systematic profile (Buolamwini, 2017) found that the IARPA Janus Benchmark A (IJB-A) (Klare et al., 2015) contained only 7.80% of faces with skin types V or VI (on the Fitzpatrick skin type scale) and again featured over 75% males. Sampling from the datasets listed here indiscriminately leads to a large proportion of images of males with lighter skin tones and upon training an image classifer often results in a biased system. Creating balanced datasets is a non-trivial task. Sourcing naturalistic images of a large number of different people is challenging. Furthermore, no matter how large the dataset is, it may still be difficult to find images that are distributed evenly across different demographic groups. Attempts have been made to improve facial classification by including gender and racial diversity. In one example, by Ryu et al. (Ryu et al., 2017), results were improved by scraping images from the web and learning facial representations from a held-out dataset with a uniform distribution across race and gender intersections.

Improving the performance of machine-learned classifiers is virtuous but there are other approaches to addressing concerns around bias. The concept of *fairness through awareness* (Dwork et al., 2012) is the principle that in order to combat bias, we need to be aware of the biases and why they occur. In complex systems, such as deep neural networks, many of the "unknowns" are unknown and need to be identified (Lakkaraju et al., 2016, 2017). Identifying and characterizing "unknowns" in a model requires a combination of *exploration* to identify regions of the model that contain failure modes and *exploitation* to sample frequently from these region in order to characterize performance. Identifying failure modes is similar to finding adversarial examples for image classifiers (Athalye et al., 2017; Tramèr et al., 2017), a subject that is of increasing interest.

One way of characterizing bias that holds promise is data simulation. Parameterized computer graphics simulators are one way of testing vision models (Veeravasarapu et al., 2015a,b, 2016; Vazquez et al., 2014; Qiu & Yuille, 2016). Generally, it has been proposed that graphics models be used for performance evaluation (Haralick, 1992). Qiu and Yuille (2016) propose applications of simulations in computer vision, including testing deep network algorithms. Recently, McDuff et al. 2018 illustrated how highly realistic simulations could be used to interrogate the performance of face detection systems. Kortlewaski et al. (2018; 2019) show that the damage of real-world dataset biases on facial recognition systems can be partially addressed by pre-training on synthetic data. However, creating high fidelity 3D assets for simulating many different facial appearances (e.g., bone structures, facial attributes, skin tones etc.) is time consuming and expensive.

Generative adversarial networks (GANs) (Goodfellow et al., 2014a) are becoming increasingly popular for synthesizing data (Shrivastava et al., 2017) and present an alternative, or complement, to graphical simulations. Generative models are trained to match the target distribution. Thus, once trained, a generative model can flexibly generate a large amount of diverse samples without the need for pre-built 3D assets. They can also be used to generate images with a set of desired properties by conditioning the model during the training stage and thus enabling generation of new samples in a

controllable way at testing time. Thus, GANs have been used for synthesizing images of faces at different ages (Yang et al., 2017a; Choi et al., 2018) or genders (Dong et al., 2017; Choi et al., 2018). However, unlike parameterized models, statistical models (such as GANs) are fallible and might also have errors themselves. For example, the model may produce an image of a man even when conditioned on a woman. To use such a model for characterizing the performance of an independent classifier it is important to first characterize the error in the image generation itself. Two motivations for using GANs are: 1) We can sample a larger number of faces than existed in the original photo dataset used to train the GAN; 2) Synthesizing face images has an advantage of preserving the privacy of individuals, thus when interrogating a commercial classifier we do not need images of "real people".

In this paper, we propose to use a characterized state-of-the-art progressive conditional generative model in order to test existing computer vision classification systems for bias, as shown in Figure 1. In particular, we train a progressive conditional generative network that allows us to create high-fidelity images of new faces with different appearances by exploiting the underlying manifold. We train the model on diverse image examples by sampling in a balanced manner from men and women from different countries. Then we characterize this generator using oracles (human judges) to identify any errors in the synthesis model. Using the conditioned synthesis model and a Bayesian search scheme we efficiently exploit and explore the parameterized space of faces, in order to find the failure cases of a set of existing commercial facial classification systems and identify biases. One advantage of this scheme is that we only need to train and characterize the performance of the generator once and can then evaluate many classifiers efficiently and systematically, with potentially many more variations of facial images than were used to train the generator.

The contributions of this paper are: (1) to present an approach for conditionally generating synthetic face images based on a curated dataset of people from different nations, (2) to show how synthetic data can be used to efficiently identify limits in existing facial classification systems, and (3) to propose a Bayesian Optimization based sampling procedure to identify these limits more efficiently. We release the nationality data, model and code to accompany the image data used in this paper (see the supplementary material).

## 2 Related Work

**Algorithmic Bias.** There is wide concern about the equitable nature of machine learned systems. Algorithmic bias can exist for several reasons and the discovery or characterization of biases is non-trivial (Hajian et al., 2016). Even if biases are not introduced maliciously they can result from explicit variables contained within a model or via variables that correlate with sensitive attributes - *indirect discrimination*. Ideally, we would minimize algorithmic bias or discrimination as much as possible, or prevent it entirely. However, this is challenging: First, algorithms can be released by third-parties who may not be acting in the public's best interest and not take the time or effort required to maximize the fairness of their models. Second, removing biases is technically challenging. For example, balancing a dataset and removing correlates with sensitive variables is very difficult, especially when learning algorithms are data hungry and the largest, accessible data sources (e.g., the Internet) are biased (Baeza-Yates, 2016).

Tools are needed to help practitioners evaluate models, especially black-box approaches. Making algorithms more transparent and increasing accountability is another approach to increasing fairness (Dwork et al., 2012; Lepri et al., 2018). A significant study (Buolamwini & Gebru, 2018) highlighted that facial classification systems were not as accurate on faces with darker skin tones and on females. This paper led to improvements in the models behind these APIs being made (Raji & Buolamwini, 2019). This illustrates how characterization of model biases can be used to advance the quality of machine learned systems.

Biases often result from unknowns within a system. Methods have been proposed to help address the discovery of unknowns in predictive models (Lakkaraju et al., 2016, 2017). In their work the search-space is partitioned into groups which can be given interpretable descriptions. Then an explore-exploit strategy is used to navigate through these groups systematically based on the feedback from an oracle (e.g., a human labeler). Bansal and Weld proposed a new class of utility models that rewarded how well the discovered *unknown unknowns* help explain a sample distribution of expected queries (Bansal & Weld, 2018). Using human oracles is labor intensive and not scalable.

We employ an explore-exploit strategy in our work, but rather than rely on a human oracle we use an image synthesis model. Using a conditioned network we provide a systematic way of interrogating a black box model by generating variations on the target images and repeatedly testing a classifier's performance. We incentivize the search algorithm to explore the parameter space of faces but also reward it for identifying failures and interrogating these regions of the space more frequently.

**Generative Adversarial Networks.** Deep generative adversarial networks have enabled considerable improvements in image generation (Goodfellow et al., 2014b; Zhang et al., 2017; Xu et al., 2018). Conditional GANs (Mirza & Osindero, 2014) allow for the addition of conditional variables, such that generation can be performed in a "controllable" way. The conditioning variables can take different forms (e.g. specific attributes or a raw image (Choi et al., 2018; Yan et al., 2016).) For facial images, this has been applied to control the gender (Dong et al., 2017), age (Yang et al., 2017a; Choi et al., 2018), hair color, skin tone and facial expressions (Choi et al., 2018) of generated faces. This allows for a level of systematic simulation via manifolds in the space of faces. Increasing the resolution of images synthesized using GANs is the focus of considerable research. Higher quality output images have been achieved by decomposing the generation process into different stages. The LR-GAN (Yang et al., 2017b) decomposes the process by generating image foregrounds and backgrounds separately. StackGAN (Zhang et al., 2017) decomposes generation stages into several steps each with greater resolution. PG-GAN (Karras et al., 2018) has shown impressive performance using a progressive training procedure starting from very low resolution images (4×4) and ending with high resolution images (1024×1024). It can produce high fidelity pictures that are often tricky to distinguish from real photos.

In this paper, we employ a progressive conditional generative adversarial model for creating photo-realistic image examples with controllable "gender" and "race". These images are then used to interrogate independent image classification systems. We model the problem as a Gaussian process (Rasmussen, 2004), sampling images from the model iteratively based on the performance of the classifiers, to efficiently discover blind-spots in the models. Empirically, we find that these examples can be used to identify biases in the image classification systems.

## 3 Approach

We propose to use a generative model to synthesize face images and then apply Bayesian Optimization to efficiently generate images that have the highest likelihood of breaking a target classifier.

**Image Generation.** To generate photo-realistic face images in a controllable way, we propose to adopt a progressively growing conditional GAN (Karras et al., 2018; Mirza & Osindero, 2014) architecture. This model is trained so as to condition the generator $G$ and discriminator $D$ on additional labels. The given condition $\theta$ could be any kinds of auxiliary information; here we use $\theta$ to specify both the race $r$ and gender $g$ of the subject in the image, i.e., $\theta = [r; g]$. During testing time, the trained $G$ should produce face images with the race and gender as specified by $\theta$.

We curated a dataset $\{x; \theta\}$ (described below), where $x$ is a face image and $\theta$ indicates the race $r$ and gender $g$ labels of $x$. To train the conditional generator, the input of the generator is a combination of a condition $\theta$ and a prior noise input $p_z(z)$; $z$ is a 100-D vector sampled from a unit normal distribution and $\theta$ is a one-hot vector that represents a unique combination of (race, gender) conditions. We concatenate $z$ and $\theta$ as the input to our model. $G$'s objective is defined by:

$$\mathcal{L}_G = -\mathbb{E}_{z,\theta}\big[\log D(G(z,\theta))\big] \tag{1}$$

The design of the discriminator $D$ is inspired by Thekumparampil et al.'s (2018) Robust Conditional GAN model which proved successful at delivering robust results. We train $D$ on two objectives: to discriminate whether the synthesized image is real or fake, and to classify the synthesized image into the correct class (e.g., race and gender). The training objective for $D$ is defined by:

$$\mathcal{L}_D = -\mathbb{E}\big[\log D(x)\big] - \mathbb{E}_{z,\theta}\big[\log D(G(z,\theta))\big] - \mathbb{E}_{z,\theta}\big[\log C(G(z,\theta))\big] \tag{2}$$

where $C$ is an N-way classifier. This classifier ($C$) is independent of the classifier being interrogated for biases. Our full learning objective is:

$$\mathcal{L}_{adv} = \min_G \max_D \mathcal{L}_G + \mathcal{L}_D \tag{3}$$

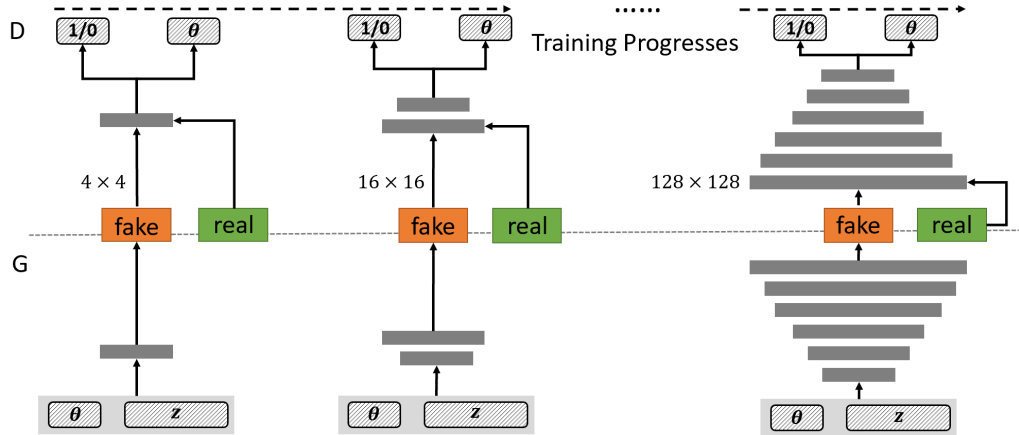

Figure 2: The training pipeline of our generative network. The input for the generator $G$ is a joint hidden representation of combined latent vectors $z$ and labels $\theta$, where $\theta$ specifies the race and gender of the face. The Discriminator $D$ is used to discriminate the generated samples from real samples (indicated by $1/0$). At the same time, $D$ forces the generated samples to be classified into the appropriate corresponding classes (indicated by $\theta$). The network is trained progressively, i.e. from a resolution of $4\times4$ pixels to $16\times16$ pixels, and eventually to $128\times128$ pixels (increasing by a factor of two).

We train the generator progressively (Karras et al., 2018) by increasing the image resolution by a power of two at each step, from $4\times4$ pixels to $128\times128$ pixels (see Figure 2). The real samples are downsampled into the corresponding resolution in each stage. The training code is included in supplementary material.

**Bayesian Optimization.** Now that we have a systematically controllable image generation model, we propose to combine this with Bayesian Optimization (Brochu et al., 2010) to explore and exploit the space of parameters $\theta$ to find errors in the target classifier. We have $\theta$ as parameters that spawn an instance of a simulation $f(\theta)$ (e.g., a synthesized face image). This instance is fed into a target image classifier to check whether the system correctly identifies $f(\theta)$. Consequently, we can define a composite function $L_c = Loss(f(\theta))$, where $Loss$ is the classification loss and reflects if the target classifier correctly handles the simulation instance generated when applying the parameters $\theta$. As an example, for gender classification the $Loss$ is 0 if the classification of gender was correct or 1 if it was incorrect. Carrying out Bayesian Optimization with $L_c$ allows us to find $\theta$ that maximizes the loss, thus discovering parameters that are likely to break the classifier we are interrogating (i.e., *exploitation*). However, we are not interested in just one instance but sets of diverse examples that would lead to misclassification. Consequently, we carry out a sequence of Bayesian Optimization tasks, where each subsequent run considers an adaptive objective function that is conditioned on examples that were discovered in the previous round. Formally, in each round of Bayesian Optimization we maximize:

$$L = (1 - \alpha)L_c + \alpha \min_{i} ||\Theta_i - \theta||. \qquad (4)$$

The above composite function is a convex combination of the misclassification cost with a term that encourages discovering new solutions $\theta$ that are diverse from the set of previously found examples $\Theta_i$. Specifically, the second term is the minimum euclidean distance of $\theta$ from the existing set and a high value of that term indicates that the example being considered is diverse from rest of the set. Intuitively, this term encourages *exploration* and prioritizes sampling a diverse set of images. In our preliminary tests, we varied the size of the set of previously found examples. We found the results were not sensitive to the size of this set and fixed it to 50 in our main experiments. Figure 1 graphically describes such a composition. The sequence of Bayesian Optimizations find a diverse set of examples by first modeling the composite function $L$ as a Gaussian Process (GP) (Rasmussen, 2004). Modeling as a GP allows us to quantify uncertainty around the predictions, which in turn is used to efficiently explore the parameter space in order to identify the spots that satisfy the search criterion. In this work, we follow the recommendations in (Snoek et al., 2012), and model the composite function via a GP with a Radial Basis Function (RBF) kernel, and use Expected Improvement (EI) as an acquisition function. A formal definition of EI is provided by Frazier Frazier (2018). The code is included in supplementary material.

| Region | Country | People | | Frames | | Generated Images | |
|---|---|---|---|---|---|---|---|
| | | M | W | M | W | M | W |
| Black | Nigerian | 81 | 28 | 768 | 467 | 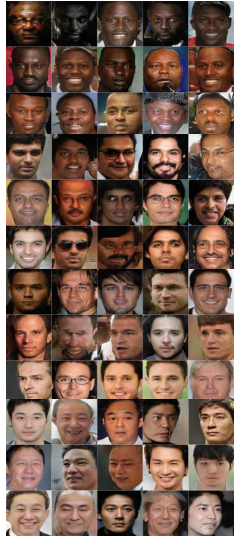    |     |
| | Kenya | 11 | 5 | 91 | 49 | | |
| | S. Africa | 136 | 102 | 1641 | 1984 | | |
| | **Total** | 228 | 135 | 2500 | 2500 | | |
| S Asian | India | 142 | 83 | 2108 | 2267 | | |
| | Sri Lanka | 1 | 2 | 11 | 7 | | |
| | Pakistan | 19 | 11 | 381 | 226 | | |
| | **Total** | 162 | 96 | 2500 | 2500 | | |
| White | Australia | 175 | 121 | 2500 | 2500 | | |
| | **Total** | 175 | 121 | 2500 | 2500 | | |
| NE Asian | Japan | 105 | 89 | 930 | 1421 | | |
| | China | 105 | 46 | 789 | 447 | | |
| | S. Korea | 29 | 12 | 464 | 251 | | |
| | Hong Kong | 36 | 28 | 317 | 381 | | |
| | **Total** | 275 | 175 | 2500 | 2500 | | |

Table 1: The number of people and images we sampled from (by country and gender) to train our generation model. Examples of generated faces for each race and gender. M = Men, W = Women.

**Data.** We use the MS-CELEB-1M (Guo et al., 2016) dataset for our experimentation. This is a large image dataset with a training set containing 100K different people and approximately 10 million images. To identify the nationalities of the people in the dataset we used the Google Search API and pulled biographic text associated with each person featured in the dataset. We then used the NLTK library to extract nationality and gender information from the biographies. Many nations have heterogeneous national and/or ethnic compositions and assuming that sampling from them at random would give consistent appearances is not well founded. Characterizing these differences is difficult, but necessary if we are to understand biases in vision classifiers. The United Nations (UN) notes that the ethnic and/or national groups of the population are dependent upon individual national circumstances and terms such as "race" and "origin" have many connotations. There is no internationally accepted criteria. Therefore, care must be taken in how we use these labels to generate images of different appearances.

To help address this we used demographic data provided by the UN that gives the national and/or ethnic statistics for each country[1] and then only sampled from countries with more homogeneous demographics. We selected four regions that have predominant and similar racial appearance groups. These group are Black (darker skin tones, Sub-Saharan African appearance), South Asian (darker skin tone, Caucasian appearance), Northeast Asian (moderate skin tone, East Asian appearance) and White (light skin tone, Caucasian appearance) and sampled from a set of countries to obtain images for each. We sampled 5,000 images (2,500 men and 2,500 women) from each region prioritizing higher resolution images (256×256) and then lower resolution images (128×128). The original raw images selected for training and the corresponding race and gender labels are included in the supplementary material. The nationality and gender labels for the complete MS-CELEB-1M training set will also be released. Table 1 shows the nations from which we sampled images and the corresponding appearance group. The number of people and images that were used in the final data are shown. It was not necessary to use all the images from every country to create a model for generating faces, and to obtain evenly distributed data over both gender and region we used this subset. Examples of the images produced by our trained model are also shown in the table. Higher resolution images can be found in the supplementary material.

## 4 Experiments and Results

**Validation of Image Generation.** Statistical generative models such as GANs are not perfect and may not always generate images that reflect the conditioned variables. Therefore, it is important to

| API | Task | All | Black | S Asian | NE Asian | White | Men | Women |
|---|---|---|---|---|---|---|---|---|
| IBM | Face Det. | 8.05 | **16.9** | **7.63** | 3.96 | 3.8 | **11.3** | 2.27 |
| | Gender Class. | 8.26 | **9.00** | 2.13 | **20.0** | 1.87 | **15.8** | 0.27 |
| SE | Face Det. | 0.13 | 0.00 | 0.00 | 0.53 | 0.00 | 0.21 | 0.00 |
| | Gender Class. | 2.84 | 3.39 | 0.74 | **5.85** | 1.38 | 5.14 | 0.00 |

Table 2: Face detection and gender classification error rates (in percentage). SE = SightEngine.

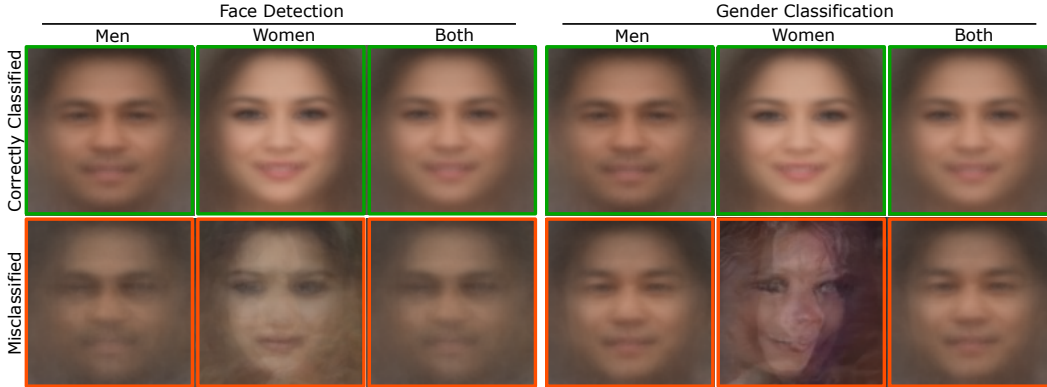

Figure 3: Mean faces for correct classifications and incorrect classifications. Notice how the skin tone for face detection failure cases is darker than for success cases. Women were very infrequently classified as men, thus the average face is not very clear.

validate the performance of the GAN that we used at producing images that represent the specified conditions (race and gender) reliably. We generated a uniform sample of 50 images, at 128×128 resolution, from each race and gender (total 50x4x2 = 400 images) and recruited five participants on MTurk to label the gender of the face in each image and the quality of the image (see Table 1 for example images). The quality of the image was labeled on a scale of 0 (no face is identifiable) to 5 (the face is indistinguishable from a photograph). Of 400 images, the gender of only seven (1.75%) images was classified differently by a majority of the labelers than the intended condition dictated. The mean quality rating of the images was 3.39 (SD=0.831) out of 5. There was no significant difference in quality between races or genders (Black Men: 3.31, Black Women: 3.24, White Men: 3.20, White Women: 3.51, S. Asian Men: 3.37, S. Asian Women: 3.40, NE Asian Men: 3.48, NE Asian Women: 3.58). In none of the images was a face considered unidentifiable.

Additionally, we computed FID scores for each region and gender, we synthesized 500 images from each region and gender (4K in total), using our conditional PG-GAN and StyleGAN (Karras et al., 2018). StyleGAN is the current state-of-the-art on face image synthesis and serves as a good reference point. The FID scores were similar across all regions: Ours (Black: M 8.10, F 8.14; White: M 8.08, F 7.70; NE Asian: M 8.01, F 8.00; S. Asian: M 8.06, F 8.10). StyleGAN (Black: M 7.70, F 7.92; White: M 7.68, F 7.8; NE Asian: M 7.76, F 7.80; S. Asian: M 7.94, F 7.66). Our model produces comparable FID scores with the state-of-the-art results. Note that StyleGAN synthesizes new images by conditioning on a real image, while our model is just conditioned on labels. The results confirm that our dataset can be used to synthesize images across each gender and region with sufficient quality and diversity.

**Classifier Interrogation.** Numerous companies offer services for face detection and gender classification from images (Microsoft, IBM, Amazon, SightEngine, Kairos, etc.). We selected two of these commercial APIs (IBM and SightEngine) to interrogate in our experiments. These are exemplars and the specific APIs used here are not the focus of our paper. Each API accepts HTTP POST requests with URLs of the images or binary image data as a parameter within the request. If a face is detected they return JSON formatted data structures with the locations of the detected faces and a prediction of the gender of the face. Details of the APIs can be found in the supplementary material.

We ran our sampling procedure for a fixed number of iterations (400) in each trial (i.e., we sampled 400 images at 128×128 resolution). In all cases the percentage of failure cases had stabilized. Table 2 shows the error rates (in %) for face detection and gender classification. Figure 3 shows the mean

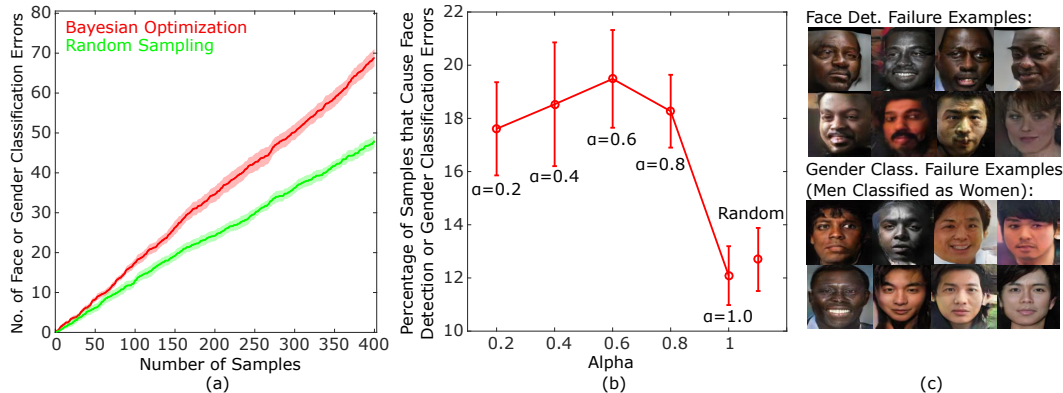

Figure 4: a) Sample efficiency of finding samples that were misclassified using random sampling and Bayesian Optimization with $\alpha$=1. Shaded regions reflect one standard deviation either side of the mean. b) Percentage of images that cause classifier failures (y-axis) as we vary the value of $\alpha$. Error bars reflect one standard deviation either side of the mean. c) Qualitative examples of failure cases.

face of images containing faces that were detected and not detected by the APIs. The skin tones illustrate that missed faces had darker skin tones and gender classification was considerably less accurate on people from NE Asia. We found men were more frequently misclassified as women than the other way around.

Next, we compare two approaches for searching our space of simulated faces for face detection and gender classification failure cases. For these analyses we used the IBM API as the target image classifier. First, we randomly sample parameters for generating face configurations and second we use Bayesian Optimization. Again, we ran the sampling for 400 iterations in each case. In the case of Bayesian Optimization, the image generation was updated dependant on the success or failure of classification of the previous image. This allows us to use an explore-exploit strategy and navigate through the facial appearance space efficiently using the feedback from the automated "oracle". Figure 4(a) shows the sample efficiency of finding face detection and gender classification failures. Figure 4(b) shows the how the percentage of the errors found varies with the value of $\alpha$ and for random sampling.

We repeated our analysis by directly sampling from the real images used to train the PG-GAN. Using BO and the GAN images allowed us to discover gender classification failures at a higher rate (16% higher) than using BO and the real images (both with $\alpha$=0.6), partially because the real image set is ultimately limited and in specific failure regions we eventually exhaust the images and have to sample elsewhere.

**Classifier Retraining.** Finally, we wanted to test whether the synthesized images could be used to improve the "weak spots" in a classifier. To do this, we trained three ResNet-50 gender classifications models. First, we sampled 14K images from our dataset (1K Black, 3K S Asian, 5K White and 5K NE Asian - this captures a typical model trained on unbalanced data). We then tested this classifier on an independent set of images and the gender classification accuracy was 81%. Following this, we then retrained the model, adding 800 synthesized failure cases discovered using our method, the retrained classifier achieved 87% accuracy on the gender classification task. Compare this to a model trained on our balanced dataset for which the accuracy was 92%. Adding the synthesized images brings performance up, close to that of a model trained on a balanced dataset.

## 5   Discussion

Bias in machine learning classifiers is problematic and often these biases may not be introduced intentionally. Regardless, biases can still propagate systemic inequalities that exist in the real-world. Yet, there are still few practical tools for helping researchers and developers mitigate bias and create well characterized classifiers. Adversarial training is a powerful tool for creating generative models that can produce highly realistic content. By using an adversarial training architecture we create a model that can be used to interrogate facial classification systems. We apply an optimal search algorithm that allows us to perform an efficient exploration of the space of faces to reveal biases from

a smaller number of samples than via a brute force method. We tested this approach on face detection and gender classification tasks and interrogated commercial APIs to demonstrate its application.

**Can our conditional GAN produce sufficiently high quality images to interrogate a classifier?** Our validation of our face GAN shows that the model is able to generate realistic face images that are reliably conditioned on race and gender. Human subjects showed very high agreement with the conditional labels for the generated images and the quality of the images were rated similarly across each race and gender. This suggests that our balanced dataset and training procedure produced a model that can generate images reliably conditioned on race and gender and of suitably equivalent quality. Examples of the generated images can be seen in Table 1 (high resolution images are available in the supplementary material). Very few of the images have very noticeable artifacts.

**How do commercial APIs perform?** Both of the commercial APIs we tested failed at significantly higher rates on images of people from the African and South Asian groups (see Table 2). For the IBM system the face detection error rate was more than four times as high on African faces as White and NE Asian faces. The face detection error rates on Black and South Asian faces were the highest, suggesting that skin tone is a key variable here. Gender classification error rates were also high for African faces but unlike face detection the performance was worst for NE Asian faces. These results suggest that gender classification performance is not only impacted by skin tone but also other characteristics of appearance. Perhaps facial hair, or lack thereof, in NE Asian photographs, men with "bangs" and make-up (see Figure 4c and supplementary material for examples of images that resulted in errors.)

**Can we efficiently sample images to find errors?** Errors are typically sparse (many APIs have a global error rate of less than 10%) and therefore simply randomly sampling images in order to identify biases is far from efficient. Using our optimization scheme we are able to identify an equivalent number of errors in significantly fewer samples (see Figure 4a). The results show that we are able to identify almost 50% more failure cases using the Bayesian sampling scheme than without. In some senses our approach can be thought of as a way of efficiently identifying adversarial examples.

**Trading off exploitation and exploration?** In our sampling procedure we have an explicit trade-off, using $\alpha$, between exploration of the underlying manifold of face images and exploitation to find the highest number of errors (see Figure 4(b)). With little exploration there is a danger that the sampling will find a local minima and continue sampling from a single region. Our results show that with $\alpha$ equal 0.6 we maximize the number of errors found. This is empirical evidence that exploration and exploration are both important. Otherwise there is risk that one might miss regions that have frequent failure cases. As the parameter space of $\theta$ grows in dimensionality our sampling procedure will become even more favorable compared to naive methods.

**Can synthesized images be used to help address errors?** Using our generative face image model we compared the performance of a gender classification model trained with unbalanced data and also trained with supplementary synthesized failure images discovered using our sampling method. The synthesized images helped successfully improve accuracy of the resulting gender classifier, suggesting that the generated images can help address problems associated with data bias. In this work we only use GANs to interrogate bias in facial classification systems across two (albeit important) dimensions: gender and race. The number of parameters could be extended with other critical dimensions, e.g., age; this could accentuate the advantages of using generative models that we have highlighted.

## 6   Conclusions

We have presented an approach applying a conditional progressive generative model for creating photo-realistic synthetic images that can be used to interrogate facial classifiers. We test commercial image classification application programming interfaces and find evidence of systematic biases in their performance. A Bayesian search algorithm allows for efficient search and characterization of these biases. Biases in vision-based systems are of wide concern especially as these system become widely deployed by industry and governments. In this work, we focus specifically on sample selection bias but it should be noted that other types of bias exist (e.g., confounding bias). Generative models are a practical tool that can be used to characterize the performance of these systems. We hope that this work can help increase the prevalence of rigorous benchmarking of commercial classifiers in the future.

## Footnotes

[1]http://data.un.org/

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
