[Supplementary Material · NIPS2019_Handling_Bias_Using_GANs_Supp_Material.pdf]

# SUPPLEMENTARY MATERIAL: Characterizing Bias in Classifiers using Generative Models

**Daniel McDuff, Yale Song and Ashish Kapoor**
Microsoft
Redmond, WA, USA
{damcduff,yalesong,akapoor}@microsoft.com

**Shuang Ma**
SUNY Buffalo
Buffalo, NY
shuangma@buffalo.edu

## 1 Image Generation Implementation Details

We provide implementation details of our conditional progressive generative adversarial model. The network architecture and parameter settings for the generator are:

$z(100)$, $\theta(8) \rightarrow$ FC(512) $\rightarrow$ Conv2D(512, 4) $\rightarrow$ LeakyReLU $\rightarrow$ Conv2D(512, 3) $\rightarrow$ LeakyReLU $\rightarrow$ Upsample $\rightarrow$ Conv2D(512, 3) $\rightarrow$ LeakyReLU $\rightarrow$ Conv2D(512, 3) $\rightarrow$ LeakyReLU $\rightarrow$ Upsample $\rightarrow$ Conv2D(512, 3) $\rightarrow$ LeakyReLU $\rightarrow$ Conv2D(512, 3) $\rightarrow$ LeakyReLU $\rightarrow$ Upsample $\rightarrow$ Conv2D(512, 3) $\rightarrow$ LeakyReLU $\rightarrow$ Conv2D(512, 3) $\rightarrow$ Upsample $\rightarrow$ Conv2D(256, 3) $\rightarrow$ LeakyReLU $\rightarrow$ Conv2D(256, 3) $\rightarrow$ LeakyReLU $\rightarrow$ Upsample $\rightarrow$ Conv2D(128, 3) $\rightarrow$ LeakyReLU $\rightarrow$ Conv2D(3, 1) $\rightarrow$ Image

The network architecture and parameter settings for the discriminator are:

Input image ($3 \times 128 \times 128$) $\rightarrow$ Conv2D(16, 1) $\rightarrow$ LeakyReLU $\rightarrow$ Conv2D(16 ,3) $\rightarrow$ LeakyReLU $\rightarrow$ Conv2D(16, 3) $\rightarrow$ LeakyReLU $\rightarrow$ Conv2D(32, 3) $\rightarrow$ Downsample $\rightarrow$ Conv2D(32, 3) $\rightarrow$ LeakyReLU $\rightarrow$ Conv2D(64, 3) $\rightarrow$ LeakyReLU $\rightarrow$ Downsample $\rightarrow$ Conv2D(128, 3) $\rightarrow$ LeakyReLU $\rightarrow$ Conv2D(256, 3) $\rightarrow$ LeakyReLU $\rightarrow$ Downsample $\rightarrow$ Conv2D(512, 3) $\rightarrow$ LeakyReLU $\rightarrow$ Conv2D(512, 4) $\rightarrow$ LeakyReLU $\rightarrow$ [FC(1); FC(8)]

We implement our learning objective using a combination of the WGAN-GP loss (Gulrajani et al., 2017) and the cross-entropy loss (for the N-way race-gender classifier). We train our conditional image generation model using the ADAM optimizer with $\alpha = 0.001$, $\beta_1 = 0$, $\beta_2 = 0.99$, and $\epsilon = 10^{-8}$; we use a batch size of 32. We alternate optimizing the generator and discriminator on a per-minibatch basis. We trained our network on 4 GeForce GTX 1080Ti GPUs for 320 epochs, which takes about 96 hours to converge. The training time increases as the size of the network grows.

## 2 Examples Images

Figure 1 shows the highest resolution images of faces generated with our conditional progressive generative adversarial network (128 x 128 pixels).

## 3 Commercial Facial Classification APIs

We tested two commercial APIs as part of our experimentation. Each API accepts HTTP POST requests with URLs of images or binary image data as a parameter within the request. If a face is detected they return JSON formatted data structures with the locations of the detected faces and a prediction of the gender of the face.

South Asian

Black

North East Asian

White

Figure 1: High resolution images of faces generated with our conditional progressive generative adversarial network (128 x 128 pixels). Left column) Men, Right column) Women.

**IBM**[1]**:** The documentation reports that the minimum pixel density is $32 \times 32$ pixels per inch, and the maximum image size is 10 MB. IBM published a statement in response to the article by Buolamwini and Gebru (2017) in which they further characterize the performance of their algorithm.[2]

**SightEngine**[3]**:** The documentation did not report minimum size or resolution requirements.

At sampling time we ran our face generation model on a single GeForce GTX 1080Ti GPU and passed images one at a time to the commercial facial APIs.

## 4   Face Classification Failures

Figure 2 shows examples of face detection failure cases of the commercial classifier (IBM). Failures most frequently occurred with darker skin tones; see Figure 2(a-g). In all these images, a face is clearly identifiable and there are few clear artifacts.

a   b   c   d   e   f   g   h   i   j   k

Figure 2: Face detection failure cases. Faces were not detected in these images. These qualitative examples reflect the quantitative results in that faces with darker skin tones were more likely to be missed than those with lighter skin tones.

Figure 3 shows examples of gender detection failure caes of the commerical classifier (IBM). Gender classification failures were most common on NE Asian men, who were misclassified as women. Notice how the failure cases occur with men who appear to have make-up (see Figure 3(c)) or who have "bangs" (see Figure 3(d-e)).

a   b   c       d   e       f   g   h   i

Figure 3: Gender classification failure cases. The gender of the subject was incorrectly classified in these images as women when in fact the images are labeled consistently by humans as men.

## Footnotes

[1]https://cloud.ibm.com/apidocs/visual-recognition

[2]http://gendershades.org/docs/ibm.pdf

[3]https://sightengine.com/docs/reference