[Reviews · NeurIPS 2019]

Reviewer 1



Originality: The task of characterizing biases in face classification systems has recently received increasing attention from researchers. While related work either use computer graphics or real-world data, here, the authors propose to use conditional GANs. Related work is mostly adequately cited, although I would recommend to also take the following related computer graphics approaches into account: - Qiu, Weichao, and Alan Yuille. "Unrealcv: Connecting computer vision to unreal engine." European Conference on Computer Vision. Springer, Cham, 2016. - Kortylewski, Adam, et al. "Analyzing and Reducing the Damage of Dataset Bias to Face Recognition With Synthetic Data." Proceedings of the IEEE Conference on Computer Vision and Pattern Recognition Workshops. 2019. - Kortylewski, Adam, et al. "Empirically analyzing the effect of dataset biases on deep face recognition systems." Proceedings of the IEEE Conference on Computer Vision and Pattern Recognition Workshops. 2018. Quality: The claims made are supported by empirical analyses, although the experimental setting is rather limited, because only two classifiers have been tested. The limitations of GANs in terms of generating realistic images have been pointed out adequately. Clarity: The quality of the manuscript is good. The paper is very well understandable. Significance: Although, the use of conditional GANs for characterizing biases in classifiers is novel, I think the manuscript in its current state is not significant enough to justify a publication at NeurIPS. The proposed work is a purely empirical analysis, no significant theoretical or technical contribution was made and the findings in this paper basically confirm those of related work (as also pointed out by the authors). Post rebuttal feedback: I would like to thank the authors for taking the effort to answer my cocnerns. The authors have shown that their approach could be used to increase the performance of biased classifiers, although there still remains a significant gap to the performance of an unbiased classifier. Nevertheless, I am willing to upgrade my review to accept, because usings GANs to interrogate biased classifiers can be potentially useful for a number of application areas.

Reviewer 2



The method proposed in the paper seems original to my knowledge. And the significance of the problem the paper trying to address is also descent in my view. I'll focus on the quality and clarity of the paper in this section. First the paper does an outstanding job on clearly stating the problem in the introduction and giving readers a detailed, comprehensive cover on the related works. But starting at the 3rd section, the paper falls short on the clarity on the method, especially the section that related to the bayesian optimization. More specifically, 1. In Eq (2), does the classifier C has anything to do with the classifier that being tested? 2. Line 173, the definition of L_c is not sufficiently clearly. The authors say "Loss is the classification loss", which is overly general. Does it has to be minimized for better classifier performance or maximized (it can go either way depending on what you use)? From the later section it seems it need to be minimized for a good classifier performance. But I would recommen clearly define what type of loss can be used here. Examples would be better. 3. The method section lacks of sufficient detail for the other researchers to fully understand the model. I know that the author provided the code in the review process but it does not reduce the importance of including sufficient detail of the method in the paper. More specicially, a). Eq(4) contains a set of perviously found examples, how large should this set be? Does the size of this set affect the performance of the proposed method? b). What is the stopping criterion for the bayeisan optimizatoin? Does it has to reach a specific number of samples? Does the loss has to be non-increasing a certain number of iterations? What would be the rationale of the choice? c). The process of bayesian optimization is very vague. The paper states that the loss is modeled as a GP. And all the other thing the authors says about the process is they use RBF kernel and EI as acquisition function. I would suggest more details on how the GP models the loss function dynamics. Related equations might also be beneficial here. d). The authors state that the image data corresponds to theta, which is a one-hot vector for genders and races. In the bayesian optimization section, does the optimization also gives one-hot vector as resulting paramters? If yes, how does the optimization algorithm constrained to generate only one-hot vectors rather than vectors in real numbers? If not, why it is reasonable to use this generated vector as input to the generator? Since the generator always takes one-hot vector as input (as shown in Eq(1)), it might perform poorly if given a vector otherwise. Pose rebuttal: thanks for the response from the authors. Their answer is clear to me. I would recommend including these explaination to the paper which will definitely improve the clarity and quality of the paper.

Reviewer 3



This paper proposes a method to discover the biases of face and gender detection systems by optimizing the conditioning variables of a GAN using bayesian optimization. Additionally, this work also releases a dataset with gender and geographic region labels. Pros: This is a well written paper Cons: The necessity of using a GAN is not well explained in the paper. As far as I understand, the objective in Eq 4 can also be optimized via querying the database created by the authors with the relevant conditioning variables theta. There is no need to generate samples conditioned on theta using a GAN. At the bare minimum, querying would use better quality images for optimizing Eq 4. In ‘Validation of Image Generation’ (Section 4) it would be helpful if authors provide a breakdown of quality ratings with both geographic region and gender (the authors do mention that there was “no significant difference” in lines 230-231, but I’d still like to see the scores for the sake of completeness.). Additionally, FID scores for each region and gender would also give further insight into the quality of images generated by the GAN. Such a search procedure is most likely an overkill for a parameter vector that has only 8 possible values (4 (geographic region) x 2 (gender); as explained in the ‘Data’ section) Post Rebuttal comments: I would like to thank the authors for their detailed response. As far as I know, GANs generally do not generate images with a greater variety than the dataset they have been trained on (as claimed by the authors on line 34 of the feedback), however I do agree with the authors that one could sample more images using a GAN. In light of their response I have upgraded the rating to accept. In the camera ready version, I would additionally like to see a comparison of the face detection failure rate of GAN + BO and Querying + BO.

[Author Response · NeurIPS 2019]

We would like to thank the reviewers for their constructive and thoughtful comments. They recognized that this is a *pressing issue* for the research community and that our work is potentially a *very important contribution* and *clearly presented*. We have made an effort to answer all their comments and will update our paper based on this rebuttal.

**R1.** We would like to highlight the contributions of this work. 1) We reveal new results, instead of simply confirming those of related work (e.g.; Buolamwini and Gebru, 2018). Prior work analyzed skin tone differences but not those across different regions. We report nuances between geographic regions including difference between South Asian and African regions, both of which feature people with darker skin tones. 2) While the individual components of our approach – cGAN, PG-GAN and Bayes sampling – already exist in the literature, we combine them in a novel way to characterize bias in classifiers. We show this approach allows for identification of 50% more failure cases than without Bayesian sampling and 16% more failure cases than without our conditional PG-GAN (see our response to R3 below). This leads to more efficient identification of bias. Furthermore, the synthesized images can then be used to improve the performance of a classifier (as described below). We will add the references suggested which are indeed relevant.

Per R1's suggestion, we performed additional experiments to show how the synthetic images can be used to improve weak spots of classifiers. Specifically. we trained three ResNet-50 gender classifications models. First, we sampled 14K images from our dataset (1K Black, 3K S Asian, 5K White and 5K NE Asian - this captures a typical model trained on unbalanced data). The classification accuracy was 81%. We then retrained this model, adding 800 synthesized failure cases discovered using our method, the retrained classifier achieved 87% accuracy. Compare this to a model trained on our balanced dataset for which the accuracy was 92%. Adding the synthesized images brings performance up, close to that of a model trained on a balanced dataset.

**R2. 1)** The classifier C in Eq (2) is independent of the classifier being tested. **2)** For the classifier being tested, the 0-1 classification loss is 0 if the classification of gender was correct or 1 if it was incorrect. We maximize this to find regions of the space in which the classifier fails the most. **3a)** In our preliminary tests, we varied the size of the set of previously found examples. We found the results were not sensitive to the size of this set and fixed it to 50 in our main experiments. **3b)** We ran the Bayesian Optimization for a fixed number of iterations (1000) in each trial. In all cases the percentage of failure cases had stabilized. As can be seen in Fig.4(a) the differences between the approaches typically become clear after several hundred iterations. **3c)** We will provide more details with the equations for Bayesian Optimization (BO). In summary, the BO routine aims to model the compositing function $L_c(\theta)$ via a Gaussian Process as $\sum_i \alpha_i$ * RBF$(\theta, \theta_i)$. Here $\theta_i$ are training data points where $L_c(\theta_i)$ has already been evaluated. Since the goal of the BO is to find $\theta$ that maximizes the loss, we choose the next point to query via Expected Improvement (EI) as the one that promises the biggest gain in utility on average. A formal definition of EI is provided in Frazier et al. 2018[1] and we will summarize it in our paper. **3d)** The BO searches a continuous eight dimensional space and outputs a real-valued vector; we apply argmax and convert it into a one-hot vector. So the generator always receives a one-hot vector.

**R3.** Our reasons for using a GAN to synthesize images are two fold: 1) We can sample a larger number of faces with a greater variability than existed in the original photo datasets; 2) Synthesizing face images has an advantage of preserving the privacy of individuals, thus when interrogating a commercial classifier we do not need images of "real people". Per R3's suggestion, we have repeated our analysis by directly sampling from the real images used to train the PG-GAN. Using BO and the GAN images allowed us to discover gender detection failures at a higher rate (16% higher) than using BO and the real images (both with $\alpha$=0.6), partially because the real image set is ultimately limited and in specific failure regions we eventually exhaust the images and have to sample elsewhere. This further supports the use of a GAN. To describe the quality for the different regions we have computed the *accuracy* of the model at producing images of the correct gender for the different regions and the *mean quality rating* (score from 0-5 in brackets): Black Men: 100% (3.31), Black Women: 98% (3.24), White Men: 100% (3.20), White Women: 100% (3.51), S. Asian Men: 100% (3.37), S. Asian Women: 100% (3.40), NE Asian Men: 88% (3.48), NE Asian Women: 100% (3.58).

To compute the FID scores for each region and gender, we synthesized 500 images from each region and gender (4K in total), using our conditional PG-GAN and StyleGAN (Karras et al., 2019). StyleGAN is the current state-of-the-art on face image synthesis (published after our paper submission) and serves as a good reference point. The FID scores were similar across all regions: Ours (Black: M 8.10, F 8.14; White: M 8.08 F 7.70; NE Asian: M 8.01 F 8.00; S Asian: M 8.06 F 8.10). StyleGAN (Black: M 7.70 F 7.92; White: M 7.68 F 7.8; NE Asian: M 7.76 F 7.80; S Asian: M 7.94 F 7.66). Our model produces comparable FID scores with the state-of-the-art results. Note that StyleGAN synthesizes new images by conditioning on a real image, while our model is just conditioned on labels. The results confirm that our dataset can be used to synthesize images across each gender and region with sufficient quality and diversity.

As this is the first work to use GANs to interrogate bias in facial classification systems we chose two particularly important dimensions to control: gender and race. The number of parameters could be extended with other critical dimensions, e.g., age; this would further accentuate the advantages of using generative models that we have highlighted.

## Footnotes

[1]Frazier, P. I. (2018). A tutorial on Bayesian optimization. arXiv preprint arXiv:1807.02811


[Meta-Review · NeurIPS 2019]

The paper proposes a method to study certain types of biases in the data-generating mode, which could, for example, translate to discrimination and unfairness in the classification setting. The reviewers agree with the importance and relevance of the proposed framework. Personally, I found the whole narrative a bit surprising, or unusual, since there is not one unique problem of “bias”, but multiple types of biases, which are largely acknowledged in the causal inference literature. For instance, (Bareinboim and Pearl, Proc. of the National Academy (PNAS), 2016) survey different types of biases such as confounding, selection, among others. In particular, if I understood the paper correctly, the authors are really discussing the mismatch between the proportion of units sampled to the study versus of the underlying population relative to certain features, which in the sciences is called (sampling) selection bias. In order to avoid readers to get confused, I would try to be more specific in the title and add a short discussion articulating the specific type of bias considered in the proposed work. Obviously, this is not to discredit in any way the technical merit of the proposed contribution.